# Mind to move: Differences in running biomechanics between sensing and intuition shod runners

Cyrille Gindre[1,2,3,4☉], Aurélien Patoz[2,5☉]*, Bastiaan Breine[2,6☉], Thibault Lussiana[1,2,3,4☉]

**1** Research and Development Department, Volodalen, Chavéria, France, **2** Research and Development Department, Volodalen SwissSportLab, Aigle, Switzerland, **3** MPFRPV, Université de Franche-Comté, Besançon, France, **4** Exercise Performance Health Innovation (EPHI) Platform, Besançon, France, **5** Institute of Sport Sciences, University of Lausanne, Lausanne, Switzerland, **6** Department of Movement and Sports Sciences, Ghent University, Ghent, Belgium

☉ These authors contributed equally to this work.
* aurelien.patoz@unil.ch

**Data Availability Statement:** The datasets for this study are freely available using the access link https://github.com/aurelienPatoz/we-run-the-way-we-are.

**Funding:** The author(s) received no specific funding for this work.

## Abstract

Delving into the complexities of embodied cognition unveils the intertwined influence of mind, body, and environment. The connection of physical activity with cognition sparks a hypothesis linking motion and personality traits. Hence, this study explored whether personality traits could be linked to biomechanical variables characterizing running forms. To do so, 80 runners completed three randomized 50-m running-trials at 3.3, 4.2, and 5m/s during which their running biomechanics [ground contact time ($t_c$), flight time ($t_f$), duty factor (DF), step frequency (SF), leg stiffness ($k_{leg}$), maximal vertical ground reaction force ($F_{max}$), and maximal leg compression of the spring during stance ($\Delta L$)] was evaluated. In addition, participants' personality traits were assessed through the Myers-Briggs Type Indicator (MBTI) test. The MBTI classifies personality traits into one of two possible categories along four axes: extraversion-introversion; sensing-intuition; thinking-feeling; and judging-perceiving. This exploratory study offers compelling evidence that personality traits, specifically sensing and intuition, are associated with distinct running biomechanics. Individuals classified as *sensing* demonstrated a more grounded running style characterized by prolonged $t_c$, shorter $t_f$, higher DF, and greater $\Delta L$ compared to *intuition* individuals ($p \leq 0.02$). Conversely, *intuition* runners exhibited a more dynamic and elastic running style with a shorter $t_c$ and higher $k_{leg}$ than their *sensing* counterparts ($p \leq 0.02$). Post-hoc tests revealed a significant difference in $t_c$ between *intuition* and *sensing* runners at all speeds ($p \leq 0.02$). According to the definition of each category provided by the MBTI, *sensing* individuals tend to focus on concrete facts and physical realities while *intuition* individuals emphasize abstract concepts and patterns of information. These results suggest that runners with sensing and intuition personality traits differ in their ability to use their lower limb structures as springs. *Intuition* runners appeared to rely more in the stretch-shortening cycle to energetically optimize their running style while *sensing* runners seemed to optimize running economy by promoting more forward progression than vertical oscillations. This study underscores the intriguing interplay between personality traits of individuals and their preferred movement patterns.

**Competing interests:** No authors have competing interests.

## Introduction

Embodied cognition, a compelling theoretical framework in cognitive science, challenges conventional notions that divorce the mind from the body [1]. This paradigm asserts a symbiotic relationship between cognitive processes and the physical body, underscoring the significance of sensory and motor experiences in shaping mental functions [2]. Unlike traditional views that confine cognition to the brain, embodied cognition recognizes the profound impact of the body's interactions with the environment on cognitive phenomena [3]. This approach emphasizes both bottom-up processes, where sensory information informs cognitive processes, and top-down influences, where higher-order cognitive functions shape our perception and interaction with the world.

Exploring the intricate dynamics of embodied cognition sheds light on the reciprocal influence between the mind, body, and the vibrant world they jointly navigate. For example, an investigation revealed that the extraverted-introverted continuum influences upright posture, with 96% of extraverted individuals maintaining an "ideally aligned" posture, while 83% of introverted individuals exhibit a kyphosis-lordosis posture [4]. Furthermore, higher levels of extraversion and conscientiousness have been linked to increased physical activity levels [5] and to faster walking speeds [6, 7], suggesting that the way individuals move may reflect their underlying personality traits. Conscientiousness, notably, has shown the capacity to mitigate the age-related decline in walking speed [6, 8]. Research has also uncovered that personality traits manifest in an individual's walking gait [9]. For instance, it was observed that greater pelvis motion in the horizontal plane during walking is associated with greater agreeableness in females, while, for males, greater thorax motion in the horizontal plane is linked with extraversion [9]. Since there were no significant distinctions in the horizontal motion of the thorax and pelvis between females and males, these correlations might be influenced by individual personalities, thus removing the influence of gender [9]. In a recent development, machine learning techniques were employed with notable accuracy to assess personality traits through the analysis of gait recorded using videos [10] or smartphone sensors [11].

Extending this line of inquiry into the realm of running, research on middle-aged male runners has identified common personality profiles and associated positive self-perception with long-term involvement in running and training [12]. Runners demonstrated heightened intelligence, creativity, self-sufficiency, sobriety, and forthrightness compared to the general population, embodying traits of introversion, shyness, and a propensity for imaginative pursuits in their personality composition [13]. Besides, a prospective study found that runners with high scores on the type A behaviour (characterized by agitation, hostility, rapid speech, and an extremely competitive nature) screening questionnaire experienced significantly more injuries, especially multiple injuries [14]. Nonetheless, limited information on the personality of recreational runners is available, primarily derived from older studies, and to the best of the author's knowledge, with no recent research on this topic. This underscores the imperative for contemporary, original research specifically addressing the personality traits of recreational runners [15].

Personality traits could be effectively classified into one of two possible categories along various axes using the Myers-Briggs Type Indicator (MBTI) test, a tool rooted in Jung's psychology [16]. Notably, there is no superior category in each MBTI axis. Additionally, recent research suggested that the duty factor (DF) plays a pivotal role in illustrating two distinct spontaneous running forms in recreational runners, i.e., runners with either low or high DF [17, 18]. DF represents the proportion of time spent in contact with the ground during a running stride and could be considered as a global variable to describe the running pattern. Both running forms (low or high DF runners) could be efficiently employed at endurance running

speeds, leading to similar running economy measures [17]. Low DF runners were shown to exhibit a shorter contact time ($t_c$), larger vertical oscillation of the center of mass during flight time ($t_f$), and more anterior (midfoot and forefoot) strike pattern, favoring elastic energy reuse. Conversely, high DF runners demonstrated a longer $t_c$, more rearfoot strike pattern, and reduced work against gravity to promote forward progression. Similarly, decreasing and increasing $t_c$ could represent two opposing yet efficient strategies for enhancing running economy [19, 20]. The first strategy involves an increase in vertical stiffness to improve running economic [19] while the second strategy posits that generating force over a longer period might be more economical [20]. Consequently, one may contemplate whether each of these running strategies could be associated with a specific personality trait category.

In the present study, we delve into the intriguing realm of embodied cognition by exploring whether an intricate connection could exist between personality traits and the spontaneous running patterns of shod runners. Indeed, the aim of this study was to explore whether the two categories of personality traits within the various MBTI axes could be linked to biomechanical variables that characterize two distinct running forms naturally embraced by individuals. This exploration should shed light on the complex relationship between the mind and motion. We hypothesized that personality traits would demonstrate association with spontaneous running patterns.

## Materials and methods

### Participants

Eighty recreational endurance runners with regular running training, 67 males (age: 29.3 ± 11.1 years, height: 178.2 ± 6.4 cm, body mass: 72.0 ± 8.5 kg, and weekly running hours: 6.4 ± 3.8 h/week) and 13 females (age: 29.8 ± 11.6 years, height: 167.2 ± 6.9 cm, body mass: 60.8 ± 9.1 kg, and weekly running hours: 8.5 ± 7.8 h/week), participated in this study. All runners identified as Caucasians. To ensure diverse participation in the study, we sought a heterogeneous panel of runners with varying training backgrounds. Consequently, participants were only mandated to run a minimum of one hour per week and maintain good self-reported general health, without any current or recent (<6 months) musculoskeletal injuries. However, nothing specific about their spontaneous running pattern such as their foot-strike pattern was required because the running pattern is assumed to be a global system with several interconnected variables [17, 18, 21, 22]. All participants completed the study on a voluntary basis. The university's institutional review board (Comité de Protection des Personnes Est 1 (CPP EST 1) approved the protocol prior to participant recruitment (ID RCB 2014-A00336-41), and the study was conducted in accordance with the latest amendments of the Declaration of Helsinki. Participants were recruited between the September 1st and November 30th of 2014. Each participant underwent two experimental sessions within one week: a running biomechanical analysis during the first session, and a personality traits assessment during the second one. All participants wore their habitual running shoes during the biomechanical analysis.

### Assessment of biomechanical variables

After providing written informed consent, participants performed a 10-min warm-up run at a self-selected speed (range: 2.5–3.5 m/s) on an indoor athletic track. Subsequently, participants completed three randomized 50-m running-trials at speeds of 3.3, 4.2, and 5 m/s starting from a standing-still position (2-min rest period between trials). These running speeds were chosen because they represent the 10-km race pace of most of endurance runners [23]. Speed was monitored using photoelectric cells (Racetime2, MicroGate, Timing and Sport, Bolzano, Italy) placed at the 20 and 40-m marks. No participants showed difficulty in running at the requested

paces. A running trial was accepted if the monitored speed was within ± 5% of the requested speed and repeated otherwise after a 2-min rest period. Less than 15% of the trials were discarded. The Optojump® photoelectric cells (MicroGate Timing and Sport, Bolzano, Italy) were used to measure $t_c$ (in ms) and $t_f$ (in ms) between the 20 and 40-m marks. The cells consist of two parallel bars which were set 1 m apart and were connected to a personal computer. One bar acts as a transmitter unit containing light emitting diodes positioned 3 mm above the ground, whereas the other bar acts as the receiver unit. When the light is interrupted by an individual's foot during running, a timer within the Optojump system records time with a precision of 1 ms (sampling frequency of 1000 Hz). This allows measuring $t_c$ as the time that the light is interrupted and $t_f$ as the time between interruptions. As for each participant, the average value over the 20-m distance was computed for $t_c$ and $t_f$ and used in what follows. The test-retest reliability of the Optojump system was demonstrated to be excellent, with low coefficients of variation (2.7%) and high intraclass correlation coefficients (range: 0.982 to 0.989) [24].

$t_c$ and $t_f$ were the basis of the other biomechanical variables computed herein. Step frequency (SF) was calculated as:

$$SF = \frac{1}{t_c + t_f},$$

and DF, reflecting the relative contribution of $t_c$ during the running stride, as [25, 26]:

$$DF = \frac{t_c}{(t_c + t_s)},$$

where $t_s$ represents the swing time, which is the time during which a foot is not in contact with the ground within a running stride, and is calculated as two consecutives $t_f$ and one $t_c$, i.e., $t_s = 2t_f + t_c$.

Furthermore, the spring-mass characteristics of the lower limb were estimated using a sine-wave model as defined by Morin, Dalleau (27). More explicitly, leg stiffness ($k_{leg}$) was calculated as:

$$k_{leg} = \frac{F_{max}}{\Delta L}.$$

$F_{max}$ represents the maximal vertical ground reaction force and was estimated using $F_{max} = mg\frac{\pi}{2}\left(\frac{t_f}{t_c} + 1\right)$, where $m$ is body mass and $g$ is gravitational acceleration [27]. $\Delta L$ represents the maximal leg compression of the spring during stance and was modeled as:

$$\Delta L = L - \sqrt{L^2 - \left(\frac{st_c}{2}\right)^2} + \Delta z,$$

where $\Delta z$ is the absolute displacement of the center of mass during stance estimated using $\Delta z = \frac{F_{max}}{m}\frac{t_c^2}{\pi^2} - g\frac{t_c^2}{8}$ [27], $s$ the running speed, and $L$ the participant's leg length, estimated as 0.53 of body height [27, 28].

## Assessment of personality traits

Based on the answers to 93 questions, the MBTI classifies personality traits into one of two possible categories along four axes: extraversion-introversion (favorite world); sensing-intuition (information processing preference); thinking-feeling (decision making); and judging-perceiving (structure). Together, these axes influence how an individual perceives a situation

and decides on a course of action. The MBTI has demonstrated excellent stability with test-retest correlations between 0.83 and 0.97 over a 4-week interval, exceeding the stability of many established trait measures, and between 0.77 and 0.84 over a 9-month interval [29]. Moreover, each dichotomy showed an agreement of 84 to 96% over 4 weeks, with a median agreement of 90% [16]. Given potential context-dependent results of the MBTI [29], the personality traits of participants were reassessed through a face-to-face meeting lasting approximately one hour, conducted by an MBTI-certified practitioner, to ensure data quality.

## Statistics

Sample size calculations determined that 80 participants were required for this study, assuming moderate effect sizes (~0.5) for biomechanical differences between MBTI axes, an α error of 0.05, and a power of 0.8 [30] and was obtained using G*Power (v3.1, available at https://www.psychologie.hhu.de/arbeitsgruppen/allgemeine-psychologie-und-arbeitspsychologie/gpower) [31]. Descriptive statistics are presented as mean ± standard deviation. Data normality and homogeneity of variance were evaluated using Kolmogorov-Smirnov and Levene's tests, respectively. Participant characteristics were compared along each MBTI axis using ANOVA and non-parametric ANOVA when data normality was not verified. Repeated-measures ANOVA (speed x MBTI axes) with Mauchly's correction for sphericity and employing Holm corrections for pair-wise post-hoc comparisons were used to investigate the effect of each MBTI axis on the biomechanical variables ($t_c$, $t_f$, DF, SF, $k_{leg}$, $F_{max}$, and $\Delta L$) while accounting for the effect of running speed. 95% confidence intervals [lower, upper] of mean differences (Δs) were calculated for each significant post-hoc comparison along the MBTI axes. Cohen's *d* effect sizes were calculated for participant characteristics along the four MBTI axes and for each significant post-hoc comparison. Effect sizes were classified as *small*, *moderate*, or *large* based on the magnitude of *d* values (0.2, 0.5, and 0.8, respectively) [32]. Statistical analysis was conducted using Jamovi (v1.6.23, available at https://www.jamovi.org), with significance set at α ≤ 0.05.

## Results

Classifications of participants along the four MBTI axes are reported in Table 1. Normality and homogeneity of variance were verified for age, height, body mass, and weekly running hours ($p \geq 0.07$; Table 1) except for age and weekly running hours which were not normally distributed ($p \leq 0.04$; Table 1). ANOVA and non-parametric ANOVA results indicated no main effect of the MBTI axes on age, height, body mass, and weekly running hours ($p \geq 0.07$; Table 1), suggesting that these characteristics were similar across each MBTI axis. Effect sizes were *small* for age, height, body mass, and weekly running hours between the categories of each MBTI axis ($|d| \leq 0.27$; Table 1), except for weekly running hours of the extraversion-introversion axis, and height and body mass of the thinking-feeling axis which were *moderate* ($0.40 \leq |d| \leq 0.66$; Table 1).

Data normality and homogeneity of variance of the biomechanical variables were all verified ($p \geq 0.07$; Table 2).

A speed x sensing-intuition axis interaction effect was observed for $t_c$ ($p = 0.02$), with no other significant interaction effects reported (other interactions: $p \geq 0.06$). Pair-wise post-hoc comparisons revealed significantly shorter $t_c$ for intuition runners compared to sensing runners at all running speeds examined ($p \leq 0.02$; Fig 1a) with *moderate* to *large* effect sizes ($0.73 \leq d \leq 1.02$).

The sensing-intuition axis reported significant differences among the biomechanical variables, leading to sensing runners showing a longer $t_c$ (Δ = 13 ms [5 ms, 21 ms]; $p = 0.002$; *small*

**Table 1. Participant characteristics (age, height, body mass, and weekly running hours) for the four Myers-Briggs Type Indicator (MBTI) axes together with their Cohen's $d$ effect size.** Significant differences ($p \leq 0.05$) are indicated in bold. Participant characteristics along each MBTI axis were compared using ANOVA and non-parametric ANOVA when data normality was not verified. Data normality and homogeneity of variance were evaluated using Kolmogorov-Smirnov and Levene's tests, respectively.

| MBTI axis | Group | Number of participants | Age (y) | Height (cm) | Body mass (kg) | Weekly running hours (h/week) |
|---|---|---|---|---|---|---|
| Extraversion-introversion | Extraversion | 37 (46%) | 30 ± 12 | 176 ± 8 | 70 ± 9 | 6 ± 4 |
| | Introversion | 43 (54%) | 29 ± 10 | 177 ± 7 | 71 ± 10 | 8 ± 5 |
| | | $d$ | -0.06 | 0.06 | 0.12 | 0.40 |
| Sensing-intuition | Sensing | 47 (59%) | 30 ± 12 | 177 ± 8 | 70 ± 10 | 6 ± 5 |
| | Intuition | 33 (41%) | 28 ± 10 | 176 ± 7 | 70 ± 9 | 7 ± 5 |
| | | $d$ | -0.15 | -0.08 | -0.02 | 0.11 |
| Thinking-feeling | Thinking | 35 (44%) | 29 ± 11 | 179 ± 7 | 72 ± 9 | 7 ± 6 |
| | Feeling | 45 (56%) | 29 ± 11 | 174 ± 8 | 68 ± 9 | 6 ± 4 |
| | | $d$ | -0.01 | -0.66 | -0.43 | -0.27 |
| Judging-perceiving | Judging | 41 (51%) | 29 ± 12 | 177 ± 8 | 70 ± 10 | 7 ± 5 |
| | Perceiving | 39 (49%) | 30 ± 10 | 175 ± 8 | 70 ± 9 | 6 ± 5 |
| | | $d$ | 0.07 | -0.24 | 0.06 | -0.11 |
| $p$-values | | ANOVA Main MBTI effect | 0.18 | 0.07 | 0.46 | 0.16 |
| | | Normality test | **<0.001** | 0.98 | 0.23 | **0.04** |
| | | Homogeneity test | 0.31 | 0.82 | 0.82 | 0.74 |

Note. Data are presented as mean ± standard deviation.

effect size; $d$ = 0.44), shorter $t_f$ ($\Delta$ = -16 ms [-22 ms, -10 ms]; $p < 0.001$; *moderate* effect size; $d$ = -0.73), higher DF ($\Delta$ = 2.1% [1.3%, 2.9%]; $p < 0.001$; *moderate* effect size; $d$ = 0.66), smaller $k_{\text{leg}}$ ($\Delta$ = -1.1 kN [-1.5 kN, -0.7 kN]; $p$ = 0.01; *moderate* effect size; $d$ = -0.68), and larger $\Delta L$ ($\Delta$ = 1.4 cm [0.7 cm, 2.1 cm]; $p$ = 0.02; *moderate* effect size; $d$ = 0.55) than intuition runners (Fig 1 and Table 3). The other three axes did not report any significant differences among the biomechanical variables ($p \geq 0.09$).

All the biomechanical variables investigated herein ($t_c$, $t_f$, DF, SF, $k_{\text{leg}}$, $F_{\text{max}}$, and $\Delta L$) reported a significant running speed effect ($p \leq 0.02$), where $t_c$ and DF decreased with increasing speed while $t_f$, SF, $k_{\text{leg}}$, $F_{\text{max}}$, and $\Delta L$ increased with increasing running speed.

## Discussion

In this exploratory study, we delved into the relationship between personality traits, as determined by the MBTI and the biomechanical characteristics of runners. On the one hand, our findings revealed distinct differences in running biomechanics between "sensing" and "intuition" runners, supporting our initial hypothesis. Sensing runners adopted a grounded running form characterized by several key biomechanical attributes. They exhibited longer $t_c$, shorter $t_f$, higher DF, and larger $\Delta L$ compared to intuition runners. In essence, sensing runners seemed to favor a more earthbound running style. Conversely, intuition runners demonstrated a more dynamic and elastic running form. They displayed shorter $t_c$ and larger $k_{\text{leg}}$ than their sensing counterparts, indicating a propensity to harness the stretch-shortening cycle and utilize their lower limb structures as efficient springs during each stride. On the other hand, no association was found between running biomechanics and the remaining three MBTI axes, contradicting our initial hypothesis.

Based on the biomechanical variables observed herein (main effect for $t_c$, $t_f$, DF, $\Delta L$, and $k_{\text{leg}}$: $p \leq 0.02$, Fig 1 and Table 1), sensing runners preferentially adopt a running form that

**Table 2. Data normality and homogeneity of variance (*p*-values) of the biomechanical variables evaluated using Kolmogorov-Smirnov and Levene's tests, respectively.** No significant difference was reported ($p > 0.05$).

| Biomechanical variable | Test (*p*) | Running speed (m/s) | | |
|---|---|---|---|---|
| | | **3.3** | **4.2** | **5.0** |
| $t_c$ | Normality | 0.73 | 0.97 | 0.90 |
| | Homogeneity | 0.70 | 0.50 | 0.29 |
| $t_f$ | Normality | 0.63 | 0.98 | 0.93 |
| | Homogeneity | 0.66 | 0.42 | 0.35 |
| DF | Normality | 0.86 | 0.85 | 0.81 |
| | Homogeneity | 0.46 | 0.17 | 0.33 |
| SF | Normality | 0.64 | 0.98 | 0.51 |
| | Homogeneity | 0.44 | 0.15 | 0.19 |
| $k_{\text{leg}}$ | Normality | 0.61 | 0.83 | 0.82 |
| | Homogeneity | 0.57 | 0.46 | 0.28 |
| $F_{\text{max}}$ | Normality | 0.64 | 0.22 | 0.45 |
| | Homogeneity | 0.80 | 0.54 | 0.79 |
| $\Delta L$ | Normality | 0.44 | 0.31 | 0.07 |
| | Homogeneity | 0.18 | 0.13 | 0.07 |

Note. contact time: $t_c$, flight time: $t_f$, duty factor: DF, step frequency: SF, leg stiffness $k_{\text{leg}}$, maximal vertical ground reaction force: $F_{\text{max}}$, and maximal leg compression of the spring during stance: $\Delta L$.

favors a larger forward displacement during $t_c$ and smaller vertical displacement of the center of mass during $t_f$ compared to intuition runners. In terms of energetics, sensing runners would optimize running economy by promoting forward progression rather than vertical oscillations of the center of mass [17]. This forward progression strategy characterizes terrestrial runners [33] as well as high DF runners [17, 18]. The linearity of the force-length relationship was shown to significantly decrease with increasing DF, suggesting a lower utilization of the spring-mass model with increasing DF [34]. These terrestrial and high DF runners were also characterized by an accentuated lower limb flexion during $t_c$ and a rearfoot strike pattern

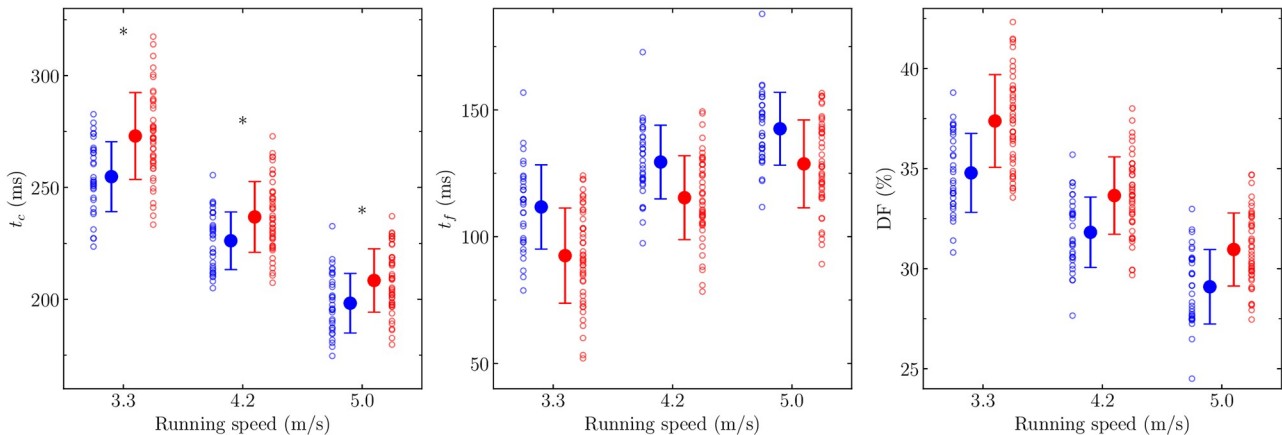

**Fig 1. Temporal characteristics of the running form for intuition and sensing runners. a**, contact time ($t_c$); **b**, flight time ($t_f$); **c**, duty factor (DF). Intuition runners (blue symbols; left side) exhibited shorter $t_c$ ($p = 0.002$), longer $t_f$ ($p < 0.001$), and lower DF ($p < 0.001$) than sensing runners (red symbols; right side). A significant speed x sensing-intuition axis interaction effect was observed for $t_c$ ($p = 0.02$). * Significantly shorter $t_c$ for intuition than sensing runners, as reported by the pair-wise post-hoc comparisons ($p \leq 0.02$). Empty circles denote the data of each participant.

**Table 3. Step frequency (SF), leg stiffness ($k_{leg}$), maximal vertical ground reaction force ($F_{max}$), and maximal leg compression of the spring during stance ($\Delta L$) for sensing and intuition runners at three tested speeds.** Significant sensing-intuition axis effects ($p \leq 0.05$) identified by the two-way repeated measures ANOVA are indicated in bold.

| Biomechanical variable | Sensing-intuition group | Running speed (m/s) | | | Sensing-intuition axis effect ($p$) |
|---|---|---|---|---|---|
| | | **3.3** | **4.2** | **5.0** | |
| SF (Hz) | Sensing | 2.7±0.1 | 2.8±0.2 | 3.0±0.2 | 0.21 |
| | Intuition | 2.7±0.1 | 2.8±0.1 | 2.9±0.1 | |
| $k_{leg}$ (kN/m) | Sensing | 8.1±1.7 | 8.1±1.6 | 8.3±1.6 | **0.01** |
| | Intuition | 9.3±1.4 | 9.0±1.4 | 9.4±1.9 | |
| $F_{max}$ (N) | Sensing | 1454±226 | 1612±231 | 1752±249 | 0.09 |
| | Intuition | 1554±192 | 1699±213 | 1860±241 | |
| $\Delta L$ (cm) | Sensing | 18±2 | 20±2 | 22±3 | **0.02** |
| | Intuition | 17±1 | 19±2 | 20±2 | |

Note. data are presented as mean ± standard deviation.

[17, 18, 33]. Sensing runners might describe their running form as: "I run very close to the ground to save as much energy as possible". These individuals, according to the definition provided by the MBTI, should pay attention to physical realities and prefer practical and specific facts, preferably something they could perceive with their physical senses [16, 35]. Hence, individuals with a more grounded running form should focus on practical facts (sensing individuals). The "physical contact" down-to-earth aspect of this personality trait seems to be reflected in both the mind and running form of sensing runners.

In contrast, intuition runners preferentially run with a larger vertical displacement of their center of mass during $t_f$ than sensing runners. The more elastic running form of intuition than sensing runners, along with their larger $k_{leg}$, suggested that the re-use of elastic energy was an inherent feature of intuition runners. These individuals were better able to use their lower limb structures as springs, representing one of the multiple functional roles of the musculoskeletal system [36]. In other words, intuition runners promote the re-use of elastic energy (spring-mass model) and rely on the stretch-shortening cycle to optimize their running economy [17]. The greater reliance on the spring-mass model was a characteristic of the aerial running form [33] as well as of low DF runners [17, 18]. These aerial and low DF runners were also characterized by an extended lower limb during $t_c$ and a forefoot/midfoot strike pattern [17, 18, 33]. Intuition runners might describe their running form as: "I spend energy to fight against gravity because I can use my leg springs to recover energy from each step". These individuals, according to the definition provided by the MBTI, should pay attention to the meaning and patterns of information, prefer abstract concepts and theories, and make unconscious connections across their disciplines of knowledge [16, 35]. Hence, individuals with a more dynamic and elastic running form should focus on abstract things (intuition individuals). While this specific study did not permit drawing causal or predictive conclusions, it highlights the fascinating interaction between an individual's personality traits and their preferred movement patterns.

Importantly, our study noted that age, height, mass, and weekly running hours did not significantly differ between sensing and intuition runners ($p \geq 0.07$), removing potential confounding variables in our analysis [37, 38]. However, it is worth noting that further investigations could explore whether differences in lower limb anatomy, such as tendon length or heel structure, might contribute to these observed biomechanical distinctions. Indeed, tendons and smaller moment arms of the Achilles tendon better support the elastic strategy than

muscles and longer moment arms [39]. In addition, larger thickness and cross-sectional area of both the Achilles tendon and plantar fascia resulted in lower DF in barefoot running [40]. Hence, such investigations might reveal thicker and slenderer lower limbs, as well as shorter heels in intuition than sensing runners. This preliminary study has raised further questions about potential interactions between body morphology, movement preferences, and personality traits. Besides, given that DF is associated with foot-strike pattern, the degree of lower limb flexion during stance, and external forces [17, 18, 34, 41], it would be valuable for future studies to investigate the connection between personality traits and these additional biomechanical variables.

It was previously demonstrated that the biomechanical characteristics of aerial and terrestrial running forms relate to feelings of pleasure-displeasure [42]. Ratings of pleasure-displeasure in runners change according to external variables, e.g., running speed. Feelings of pleasure are positively impacted in runners in situations where they are more biomechanically efficient, i.e., individuals with shorter $t_c$ and longer $t_f$ prefer running at faster speeds. As locomotion performance reflects trade-offs between different aspects of an individual's biomechanics and environmental conditions [43], and that these aspects are linked with feelings of pleasure-displeasure, we could expect that intuition and sensing runners would take more pleasure at faster and slower running speeds, respectively. This assumption aligns with the MBTI description of both personality traits, where intuitive individuals are described as people living in the fast world of future possibilities, and sensing individuals as people living in the slow world of concrete things [16]. With such an integrative perspective that considers an individual's movement patterns and environmental conditions, we can speculate that sensing and intuition runners would prefer different environments, supporting the theoretical framework of embodied cognition [2, 3]. For instance, intuition runners may lean towards shorter and faster running events, opt for harder running surfaces, and favor more minimalist running shoes, whereas sensing runners may gravitate towards longer and slower running events, softer surfaces, and opt for more cushioned running shoes, reflecting their potential connection to DF and, consequently, the intuition-sensing personality. The assumption about the choice of running shoes is in line with previous observations that runners who have attempted barefoot running tend to be more open and less conscientious than shod runners [44]. Future work may further explore the interaction between personality traits, running biomechanics, and several environmental variables, including ground surface, running speed, and running footwear.

Notwithstanding, understanding the connections between personality traits and movement holds potential public health implications. Indeed, tailoring physical interventions through suitable exercises and instructions could mitigate non-adherence [45] and variability in responses [46] to a running training program in the context of a modern sedentary lifestyle. The disparities in running biomechanics associated with sensing and intuition personality traits might result in distinct injury locations or different underlying causes for a given injury. This suggests the need for tailored rehabilitation treatments, as previously advocated [47]. These observations partially align with findings from a prospective study, indicating that runners characterized by agitation, hostility, rapid speech, and an extremely competitive nature (Type A behavior) encountered significantly more injuries, particularly multiple injuries [14].

A few limitations to the present study exist. First, no causal or predictive conclusions could be drawn using this specific study's design, but this study provides valuable information about personality traits and running forms. Then, even though runners were shown to demonstrate their most valid biomechanical running characteristics at their preferred running speed [48], biomechanical variables were evaluated at fixed running speeds to allow us comparing these variables between individuals. Besides, the MBTI validity has been questioned [49] and is regarded as a controversial approach [50], with psychometric limitations [51, 52].

Nevertheless, this tool is still the most widely used personality assessment in the world [29, 35]. Moreover, MBTI correlates well with the Neuroticism, Extraversion, Openness (NEO) Personality Inventory, another widely used personality assessment tool that examines the Big Five personality traits [53, 54] and MBTI has been utilized, though several decades ago, to assess personality traits in middle-age male runners [12]. The MBTI was preferred over the Big Five in the present study due to its nuanced nature. The MBTI assigns a personality trait among two distinct categories for each axis, as opposed to the Big Five, which merely indicates the absence or presence of a given personality trait. As researchers, we assert that the Big Five tends to involve value judgments, whereas the MBTI assigns one of two possible personality traits to each axis without implying superiority for either. Next, several factors, such as emotion, mood, or facial expression, which were not measured herein, might have partly confounded the results of the present study. For instance, Williams, Exell [55] reported that sadness might increase running asymmetry while anger might facilitate symmetry and Brick, McElhinney [56] showed that oxygen consumption was lower when smiling than frowning during running and perceived effort was higher when frowning than smiling. However, to the best of authors knowledge, there was no direct scientific evidence that these factors could influence the biomechanical variables measured herein. Moreover, the present study did not account for sex distinctions. Despite utilizing a relatively large sample size ($n = 80$), the decision was made not to differentiate between males ($n = 67$) and females ($n = 13$) to maintain simplicity and ensure an easily comprehensible manuscript, additionally given that separating the genders would have compromised statistical power. Nevertheless, future investigations should prioritize exploring the influence of sex when analyzing the connection between personality traits and running patterns, considering the demonstrated but subtle differences in personality types between males and females [57]. Furthermore, participants wore their own running shoes during testing, which could be confounding our results. Given that differences in footwear characteristics can underpin differences in running biomechanics [58–62], using a standardized shoe might have led to different study outcomes in terms of running biomechanics. Nonetheless, recreational runners are more comfortable wearing their own shoes [63], and show individual responses to novel footwear [63, 64] and cushioning properties [65]. Finally, this study did not measure the foot-strike pattern of participants, despite existing biomechanical variations reported among different patterns [66, 67]. Notably, forefoot and midfoot strikers exhibited significantly shorter contact times $t_c$ compared to heel strikers [68]. However, it's crucial to recognize that the foot-strike pattern is just one element within the broader running pattern, encompassing various interconnected variables [17, 18, 21, 22]. Considering this, runners with a more grounded running form, and associated with *sensing* personality trait, should exhibit a more rearfoot strike pattern because of the longer $t_c$, while those with a more dynamic and elastic form, often associated with *intuition* personality trait, should demonstrate a more forefoot/midfoot strike pattern due to the shorter $t_c$. Nevertheless, this statement requires validation through future research.

## Conclusions

This exploratory study offers compelling evidence that personality traits, specifically sensing and intuition, are associated with distinct running biomechanics. Sensing runners, who pay attention to physical realities and prefer practical and specific facts, tend to adopt a more grounded running form associated with longer $t_c$, shorter $t_f$, higher DF, and larger $\Delta L$ than intuition runners. On the contrary, intuition runners, who prefer abstract concepts and theories, and make unconscious connections across their disciplines of knowledge, tend to opt for a more dynamic and elastic running form with shorter $t_c$ and larger $k_{leg}$ than sensing runners.

## Supporting information

**S1 File. Personal protection committee EST I.**
(PDF)

## Acknowledgments

We thank Dr. Jean-Denis Rouillon and Prof. Laurent Mourot (University of Franche-Comté) for initiating this study. We also thank Dr. Kim Hébert-Losier (University of Waikato, New Zealand) for useful discussions and comments on the manuscript. We thank Stephanie Giordano Assante (MBTI certified practitioner) for the assessment of the personality traits of participants. We are grateful to the many volunteer runners who participated in this experiment.

## Author Contributions

**Conceptualization:** Cyrille Gindre, Thibault Lussiana.

**Data curation:** Aurélien Patoz, Bastiaan Breine, Thibault Lussiana.

**Formal analysis:** Aurélien Patoz, Bastiaan Breine, Thibault Lussiana.

**Writing – original draft:** Aurélien Patoz, Thibault Lussiana.

**Writing – review & editing:** Cyrille Gindre, Aurélien Patoz, Bastiaan Breine, Thibault Lussiana.

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
