## [Decision Letter · Decision Letter 0]

26 Nov 2023

PONE-D-23-34932We run the way we are: exploring the connection between personality traits and running biomechanicsPLOS ONE

Dear Dr. patoz,

Thank you for submitting your manuscript to PLOS ONE. After careful consideration, we feel that it has merit but does not fully meet PLOS ONE’s publication criteria as it currently stands. Therefore, we invite you to submit a revised version of the manuscript that addresses the points raised during the review process.

We look forward to receiving your revised manuscript.

Kind regards,

Yaodong Gu

Academic Editor

PLOS ONE

Journal Requirements:

Additional Editor Comments:

Discussion shall have more evidence supporting the main findings.

Reviewers' comments:

Reviewer's Responses to Questions

**Comments to the Author**

1. Is the manuscript technically sound, and do the data support the conclusions?

Reviewer #1: Partly

Reviewer #2: Yes

Reviewer #3: Partly

2. Has the statistical analysis been performed appropriately and rigorously? 

Reviewer #1: No

Reviewer #2: Yes

Reviewer #3: No

3. Have the authors made all data underlying the findings in their manuscript fully available?

Reviewer #1: No

Reviewer #2: Yes

Reviewer #3: Yes

4. Is the manuscript presented in an intelligible fashion and written in standard English?

Reviewer #1: Yes

Reviewer #2: Yes

Reviewer #3: No

5. Review Comments to the Author

Reviewer #1: Although the writing of the article is almost good, but the writing of the results has been done poorly. it is suggested

1- The results of the normality of the data should be reported in the text in the form of a table

2- The demographic characteristics of the participants should be mentioned and reported in the table

3- The entry and exit criteria of people participating in the study should be given.

4- I suggest putting a diagram of how to assessment running speed in the text.

In general, the two parts of the study method and the results need a general review.

Reviewer #2: The study aimed to explore the potential association between biomechanical variables and personality traits assessed through the Myers-Briggs Type Indicator test in a cohort of 80 runners. Although it is a very interesting study, there are several significant concerns that need to be addressed.

1. There are several areas in the Abstract section that require improvement. 1) please add some background information to better emphasize the novelty and significance of the research. 2) there is no description about the research methods here, which makes it hard to understand the results of this study. 3) The conclusion should be further strengthened based on the main findings of the study. The study conclusion should be concise and indicate the most important findings of the research; 6) it would be beneficial to improve the quality of the keywords used to make it easier for researchers and other interested parties to locate information relevant to the research topic.

2. In the Introduction section, the authors described their objective to explores the potential association between biomechanical variables and personality traits. However, the explanation of the research gap was insufficient, and the novelty of the study was not clearly communicated. Merely stating that no research has been conducted on this topic does not necessarily indicate the significance of the study. It is essential to explain what new insights this study can bring to the existing knowledge in this field and how it is relevant to practical situations. In addition, please add the corresponding hypotheses at the end of this session.

3. The Methods section of the study provides limited information on the participants and data analysis. To improve the clarity and comprehensiveness of this section, the following suggestions are recommended. 1) the inclusion criteria were relatively broad, for example, how about the running strike patterns of these participants? 2) “3.3, 4.2, 5 m/s”, why did the authors choose these running speeds? To represent low, moderate, and high running speeds? For what purposes? 3) did all participants wear the same running shoes during the test? 4) how did the optical measurement system capture the motion and calculate the corresponding parameters? 5) “a 20-m distance”, I don’t think that 20-m distance is enough to reach a stable 5m/s running speed. 6) this study explores the potential association between biomechanical variables and personality traits, thus how did the authors avoid the influence of other conditions? for example, the mood of the participants may have some influence on the gait features.

4. For the results of this study, 1) it is suggested that a table could be used to clearly present the basic information of the participants and the results of the personality traits assessment. 2) it is suggested to further calculate the effect size of these results.

5. Discussion, it would be interesting for future study to further investigate the gender influence during this test.

6. The conclusion should be further strengthened based on the main findings of the study.

Reviewer #3: General comments

In the study titled "We Run the Way We Are: Exploring the Connection Between Personality Traits and Running Biomechanics," 80 runners were included, and the relationship between biomechanical variables and personality traits, using the Myers-Briggs Type Indicator, was determined. Runners were classified into two types: "Sensing" and "Intuition." This study is intriguing as it investigates the association between biomechanics and psychological variables, offering potential insights into human running. However, the manuscript lacks a clear rationale for the study throughout. Several significant improvements are needed to enhance the study, and it is advisable to have a native speaker edit the manuscript.

Specific comments (**line numbers are based on the PDF copy)

Title

I suggest revising the title to "Differences in Running Biomechanics between Sensing and Intuition Shod Runners."

Introduction

1. This section needs strengthening to provide a compelling reason for conducting the study. The authors should emphasize the significance of the current study and how the findings can improve running performance or prevent running injuries.

2. Establish a logical connection between biomechanics and psychology variables rather than introducing measurements of running patterns and the Myers-Briggs Type Indicator.

3. Rewrite the sentence on lines 56-58 for better clarity.

4. Summarize findings from previous studies between different motion and personality traits (lines 63-71).

Methods

1. Clarify if all runners are short-distance runners and provide a clear definition of training hours.

2. Explain why participants were recruited in 2014, and address concerns about the study not being published in the last nine years.

3. Justify the use of 3.3, 4.2, and 5 m/s running speeds (lines 150-151).

4. Provide more details on the lab setting, data reduction, and motion capture methods.

Results

1. Address any differences in population characteristics between "Sensing" and "Intuition" groups.

2. Double-check values in Table 1, specifically △L.

3. Report 95% confidence intervals of mean differences in pair-wise comparisons.

Discussion:

1. Include more evidence supporting the findings, considering factors like landing patterns, running cadence, surfaces, and footwear.

Conclusion:

Refine this section for clarity.

References:

Update recent references.

6. PLOS authors have the option to publish the peer review history of their article (what does this mean?). If published, this will include your full peer review and any attached files.

Reviewer #1: No

Reviewer #2: No

Reviewer #3: No

---

## [Decision Letter · Decision Letter 1]

18 Jan 2024

PONE-D-23-34932R1We run the way we are: differences in running biomechanics between Sensing and Intuition shod runnersPLOS ONE

Dear Dr. patoz,

Thank you for submitting your manuscript to PLOS ONE. After careful consideration, we feel that it has merit but does not fully meet PLOS ONE’s publication criteria as it currently stands. Therefore, we invite you to submit a revised version of the manuscript that addresses the points raised during the review process.

We look forward to receiving your revised manuscript.

Kind regards,

Yaodong Gu

Academic Editor

PLOS ONE

Additional Editor Comments:

The limitation of this study shall be further discussed.

Reviewers' comments:

Reviewer's Responses to Questions

**Comments to the Author**

1. If the authors have adequately addressed your comments raised in a previous round of review and you feel that this manuscript is now acceptable for publication, you may indicate that here to bypass the “Comments to the Author” section, enter your conflict of interest statement in the “Confidential to Editor” section, and submit your "Accept" recommendation.

Reviewer #2: (No Response)

Reviewer #4: (No Response)

2. Is the manuscript technically sound, and do the data support the conclusions?

Reviewer #2: Yes

Reviewer #4: Partly

3. Has the statistical analysis been performed appropriately and rigorously? 

Reviewer #2: Yes

Reviewer #4: Yes

4. Have the authors made all data underlying the findings in their manuscript fully available?

Reviewer #2: Yes

Reviewer #4: Yes

5. Is the manuscript presented in an intelligible fashion and written in standard English?

Reviewer #2: Yes

Reviewer #4: No

6. Review Comments to the Author

Reviewer #2: Despite the extensive revisions made by the author based on previous comments, there are still issues that need to be addressed. I recommend that the author make further modifications based on the following suggestions.

1. The author did not indicate the influence of speed on the results in the abstract.

2. In terms of previous comment “1) the inclusion criteria were relatively broad, for example, how about the running strike patterns of these participants?”, the running strike patterns would have a significant influence on the lower limb biomechanics of running, such as vertical loading rates of GRF, foot and knee angle, and so on (https://www.jospt.org/doi/10.2519/jospt.2015.6019). How does the author eliminate this influence? Or has the author considered these effects?

3. “Given that differences in footwear characteristics can underpin differences in running biomechanics (52), using a standardized shoe might have led to different study outcomes in terms of running biomechanics.”, it is suggested that the author cite recent relevant studies to further elaborate on this limitation. I leave to the authors some bibliography to complement this information (https://doi.org/10.3390/bioengineering9110607;
https://doi.org/10.1016/j.jbiomech.2023.111597 ).

Reviewer #4: General Comment：

This study aims to contrast the biomechanical variances observed in runners with distinct personality traits, specifically, the sensate type and the intuitional type, during running exercises. The findings illustrated distinct running biomechanics associated with varying personality traits, particularly those related to perception and intuition. The authors' investigation into the intricate relationship between individual personality traits and running patterns is genuinely commendable, yielding substantial results. However, the authors may have overlooked the robustness of the experiment, given its completion in 2014. Additionally, certain aspects of the logical flow in the manuscript seemed somewhat stretched, lacking a plausible explanation, thus falling short of our journal's criteria concerning experimental processes and textual logic. As a result, regrettably, the reviewers deemed this study unsuitable for publication in our journal.

Specific comments are shown below:

Abstract

Minor：

[Comment 1] Line50-53: In contrast to individuals with intuition traits, those with sensing traits exhibited significant differences (p≤0.02) in ground contact time, flight time, duty factor, and maximal leg compression. This finding is somewhat surprising.

[Comment 2] Line56-57: Sensing individuals tend to focus on concrete facts and physical realities while intuition individuals emphasize abstract concepts and patterns of information” How did the author measure this particular result?

1.Introduction:

Major：

The introduction section suggests a motivation for the study based on the scarcity of recent personality surveys on recreational runners, leading the authors to explore the relationship between personality traits and running biomechanics. However, this motivation may not resonate strongly with readers and reviewers. The reviewers recommend that the authors enhance the practical benefits of the study to establish its relevance more effectively.

Minor：

[Comment 1] Line88-91: For the variations in gait characteristics between males and females, is this due to individual personalities or to gender?

[Comment 2] Line101-103: The authors were advised to incorporate more recent studies in the field, preferably from the last 5 years, to better capture the contemporary relationship between an individual's personality and their choice of sport. Utilizing more recent literature would enhance the relevance and applicability of the findings to current circumstances.

[Comment 3] Line114-116: Why did the authors choose the MBTI test instead of the Big Five Inventory? As far as the reviewers realized, the reliability and validity of the MBTI has many psychometric limitations and is therefore not recommended for widespread academic use. Regarding “Boyle G J. Myers‐Briggs type indicator (MBTI): some psychometric limitations[J]. Australian Psychologist, 1995, 30(1): 71-74.”，“ Stein R, Swan A B. Evaluating the validity of Myers‐Briggs Type Indicator theory: A teaching tool and window into intuitive psychology[J]. Social and Personality Psychology Compass, 2019, 13(2): e12434.”

[Comment 4] Line121-123: Generally, there is a possibility that the extended contact time observed in athletes could be a strategy employed to conserve energy during running. Regarding :Lempke A F D J, Hunt D L, Willwerth S B, et al. Biomechanical changes identified during a marathon race among high-school aged runners[J]. Gait & Posture, 2024, 108: 44-49. As the author discusses commonly employed running skills among athletes, it remains unclear how the author correlates these skills with athlete personality traits.

[Comment 5] Line125-127: The reviewer suggests that the author consider slightly restructuring the statement to make it more fluent: In the present study, we delve into the intriguing realm of embodied cognition by exploring whether an intricate connection could exist between personality traits and the spontaneous running patterns of shod runners.

2. Materials and Methods

Minor：

[Comment 1] Line140: What is the author's purpose in mentioning "white" Caucasians here?

[Comment 2] Line140-142: In the description of the participants above, the average amount of time the subjects ran per week was approximately 6 hours for males and 8 hours for females. Here the authors only asked the participants to perform at least 1 hour per week, does the intensity of the training meet the training requirements of the participants? What is the authors' basis for setting 1 hour?

[Comment 3] Line142-143: Why were the authors not specific in terms of participants' foot impact patterns? As mentioned in previous studies, there may be a greater correlation between foot impact pattern, running economy, and ground contact time（Hayes P, Caplan N. Foot strike patterns and ground contact times during high-calibre middle-distance races[J]. Journal of sports sciences, 2012, 30(12): 1275-1283; Di Michele R, Merni F. The concurrent effects of strike pattern and ground-contact time on running economy[J]. Journal of science and medicine in sport, 2014, 17(4): 414-418; Almeida M O, Davis I S, Lopes A D. Biomechanical differences of foot-strike patterns during running: a systematic review with meta-analysis[J]. Journal of Orthopaedic & Sports Physical Therapy, 2015, 45(10): 738-755.）Therefore, the reviewers recommended that the authors should screen participants on their foot strike patterns.

[Comment 4] Line 143-145: As a human experiment, all participants were entitled to be informed about the content of the experiment in which they were participating.

[Comment 5] Line 147-149: As the authors acknowledged, this experiment was conducted in 2014. Given the recent surge in long-distance running activities, such as marathons and trail running, involving individuals with diverse personalities, what contemporary relevance and application can be attributed to the authors' research a decade later?

[Comment 6] Line151-152: Why did the authors not make any special requirements regarding the participants' running shoes? Previous studies have shown that different running shoes, especially for longer distances, provide an essential role in running economy and individualized performance. Roca-Dols A, Losa-Iglesias M E, Sánchez-Gómez R, et al. Effect of the cushioning running shoes in ground contact time of phases of gait[J]. Journal of the mechanical behavior of biomedical materials, 2018, 88: 196-200; De Wit B, De Clercq D, Lenoir M. The effect of varying midsole hardness on impact forces and foot motion during foot contact in running[J]. Journal of applied biomechanics, 1995, 11(4): 395-406. Hence, if participants had utilized distinct running shoes, it would have evidently introduced additional variables during the experiment.

[Comment 7] Line158-160: What is the author's criteria for judging these three speeds as "low, medium, and fast"?

[Comment 8] Line 186-187: What is the basis for the authors' use of this formula for calculating the participant's maximum vertical ground reaction force?

[Comment 9] Line190: The article lacks clarification on the model and device used by the authors to capture the subject's center of mass.

[Comment 10] Line191-192: What is the basis for the 0.53 value? Please cite relevant literature and adapt it to this study.

[Comment 11] Line206-207: The author is kindly asked to add an explanation for the first occurrence of the abbreviation (NEO)

[Comment 12] Line209-210: The study on runners referenced by the authors was conducted in 1991, a notably different era influenced by changes in running technology and societal shifts. Additionally, it remains unclear if there is a correlation between enhancing team cohesion, understanding team dynamics, and the findings of this study.

[Comment 13] Line210-213: Can the authors elaborate on the connection between the personalities of these students and the recreational runners recruited for this study?

[Comment 14] Line216-218: The authors were asked to add tools for sample calculations.

3.Results

Minor：

[Comment 1] The authors are requested to enhance the clarity of Figure 1 and 2, furthermore, the authors do not seem to have mentioned in the methods how the standardization of the phases will be done, which the authors should be kindly requested to add.

[Comment 2] Line288: In reference to ∆ in Table 3, the findings of the authors indicate that the most significant alteration in ∆ transpired at higher velocities. In practical training scenarios, ∆ appears to exhibit a declining tendency with escalating running speeds. Regarding: Jaén-Carrillo D, Roche-Seruendo L E, Felton L, et al. Stiffness in running: a narrative integrative review[J]. Strength & Conditioning Journal, 2021, 43(2): 104-115.）

4. Discussion

Minor：

[Comment 1] Line324-327: What does the author base this conclusion on?

[Comment 2] Line330-334: The author is kindly asked to redescribe the sentence to make it clearer, as well as to correct its tenses.

[Comment 3]Line341-342: What indicators are the authors basing their observations on?

[Comment 4] Line344: Why does Figure 2 appear here?

5. Conclusions

Major：

Reviewers anticipate authors to succinctly summarize and expound upon the tangible outcomes of this study, eschewing vague remarks

7. PLOS authors have the option to publish the peer review history of their article (what does this mean?). If published, this will include your full peer review and any attached files.

Reviewer #2: No

Reviewer #4: No

---

## [Decision Letter · Decision Letter 2]

19 Feb 2024

PONE-D-23-34932R2Mind to move: differences in running biomechanics between Sensing and Intuition shod runnersPLOS ONE

Dear Dr. patoz,

Thank you for submitting your manuscript to PLOS ONE. After careful consideration, we feel that it has merit but does not fully meet PLOS ONE’s publication criteria as it currently stands. Therefore, we invite you to submit a revised version of the manuscript that addresses the points raised during the review process.

We look forward to receiving your revised manuscript.

Kind regards,

Yaodong Gu

Academic Editor

PLOS ONE

Journal Requirements:

Reviewers' comments:

Reviewer's Responses to Questions

**Comments to the Author**

1. If the authors have adequately addressed your comments raised in a previous round of review and you feel that this manuscript is now acceptable for publication, you may indicate that here to bypass the “Comments to the Author” section, enter your conflict of interest statement in the “Confidential to Editor” section, and submit your "Accept" recommendation.

Reviewer #2: All comments have been addressed

Reviewer #4: (No Response)

2. Is the manuscript technically sound, and do the data support the conclusions?

Reviewer #2: (No Response)

Reviewer #4: Yes

3. Has the statistical analysis been performed appropriately and rigorously? 

Reviewer #2: (No Response)

Reviewer #4: Yes

4. Have the authors made all data underlying the findings in their manuscript fully available?

Reviewer #2: (No Response)

Reviewer #4: Yes

5. Is the manuscript presented in an intelligible fashion and written in standard English?

Reviewer #2: (No Response)

Reviewer #4: Yes

6. Review Comments to the Author

Reviewer #2: (No Response)

Reviewer #4: General Comment：

The reviewers express delight in witnessing the meticulous adaptation and revision of the article's proposal by the authors, leading to enhanced logical coherence and methodological rigor. This research article is commended for its innovative propositions, and the reviewer encourages the author to strive for the utmost professionalism and perfection in presenting their work.

Specific comments are shown below:

Abstract

Major: Authors are kindly requested to furnish the essential abbreviations in the abstract section, with a particular emphasis on the depiction of methods, results, and conclusions.

1.Introduction:

Minor：

[Comment 1] Lines 100-103: The reviewer raised concerns regarding the authors' mention of an association between population characteristics and shoeing habits. As per the methodology described in the article, all subjects had their own footwear, but no survey of footwear was conducted. Therefore, the description of the context of the study lacks real relevance to the study itself, and the reviewer recommends its deletion.

[Comment 2] Lines 124-128: In this section, the authors discuss the effect of heel strike patterns on running performance. Obviously, this contradicts the authors' response that the spontaneous running pattern is a whole system with multiple interrelated variables and is therefore not entirely dependent on the walking pattern.

2.Patients and Methods

Minor：

[Comment 1]Lines 151-153: In the author's response, the author objectively argues that a training regimen of at least one hour per week is sufficient for successful completion of the program. The reviewer would like to emphasize that such a training requirement is clearly too low to meet the normal training needs of the subjects. Additionally, the concept seems relatively abstract, and it is possible that some subjects completed more than 8 hours of training despite the experiment's demands. The reviewers would have appreciated the authors specifying the exact training intensity of the subjects during the week prior to the experiment.

7. PLOS authors have the option to publish the peer review history of their article (what does this mean?). If published, this will include your full peer review and any attached files.

Reviewer #2: No

Reviewer #4: No

---

## [Decision Letter · Decision Letter 3]

23 Feb 2024

Mind to move: differences in running biomechanics between Sensing and Intuition shod runners

PONE-D-23-34932R3

Dear Dr. patoz,

We’re pleased to inform you that your manuscript has been judged scientifically suitable for publication and will be formally accepted for publication once it meets all outstanding technical requirements.

Kind regards,

Yaodong Gu

Academic Editor

PLOS ONE

Additional Editor Comments (optional):

Well done!

Reviewers' comments:

Reviewer's Responses to Questions

**Comments to the Author**

1. If the authors have adequately addressed your comments raised in a previous round of review and you feel that this manuscript is now acceptable for publication, you may indicate that here to bypass the “Comments to the Author” section, enter your conflict of interest statement in the “Confidential to Editor” section, and submit your "Accept" recommendation.

Reviewer #4: All comments have been addressed

2. Is the manuscript technically sound, and do the data support the conclusions?

Reviewer #4: Yes

3. Has the statistical analysis been performed appropriately and rigorously? 

Reviewer #4: Yes

4. Have the authors made all data underlying the findings in their manuscript fully available?

Reviewer #4: No

5. Is the manuscript presented in an intelligible fashion and written in standard English?

Reviewer #4: Yes

6. Review Comments to the Author

Reviewer #4: We sincerely thank the authors for responding positively to all of the reviewers' comments. All comments were thoroughly explained and revised, significantly enhancing the quality and logic of the article. The reviewers have agreed to publish the article in this journal

7. PLOS authors have the option to publish the peer review history of their article (what does this mean?). If published, this will include your full peer review and any attached files.

Reviewer #4: No

---

## [Editor Report · Acceptance letter]

13 Mar 2024

PONE-D-23-34932R3 

PLOS ONE

Dear Dr. Patoz, 

I'm pleased to inform you that your manuscript has been deemed suitable for publication in PLOS ONE. Congratulations! Your manuscript is now being handed over to our production team.

Kind regards, 

on behalf of

Professor Yaodong Gu 

Academic Editor

PLOS ONE